# The effect of perceived interracial competition on psychological outcomes

**Jonathan Gordils**[ID]*, **Andrew J. Elliot, Jeremy P. Jamieson**

Department of Psychology, University of Rochester, Rochester, New York, United States of America

* jonathan.gordils@rochester.edu

## Abstract

There remains a dearth of research on causal roles of perceived interracial competition on psychological outcomes. Towards this end, this research experimentally manipulated perceptions of group-level competition between Black and White individuals in the U.S. and tested for effects on negative psychological outcomes. In Study 1 ($N = 899$), participants assigned to the high interracial competition condition (HRC) reported perceiving more discrimination, behavioral avoidance, intergroup anxiety, and interracial mistrust relative to low interracial competition (LRC) participants. Study 2 –a preregistered replication and extension—specifically recruited similar numbers of only Black and White participants ($N = 1,823$). Consistent with Study 1, Black and White participants in the HRC condition reported more discrimination, avoidance, anxiety, and mistrust. Main effects for race also emerged: Black participants perceived more interracial competition and negative outcomes. Racial income inequality moderated effects; competition effects were stronger in areas with higher levels of inequality. Implications for theory development are discussed.

**Data Availability Statement:** All relevant data are within the manuscript and its Supporting information files.

**Funding:** The author(s) received no specific funding for this work.

## Introduction

Competition—zero-sum vying for valued resources [1, 2]–is ubiquitous [3, 4]. From playing low-stakes games to striving for college admission or placing an offer on a home, competition is pervasive and has a powerful influence on numerous psychological, behavioral, and health outcomes [5, 6]. Competition is not only prevalent at the individual level, such as job applicants contending for the same position, but also frequently manifests between social groups [7]. That is, social groups, including racial, gender, or religious groups to name a few, compete with (or are perceived as competing with) other groups for limited societal resources. Regarding interracial competition, specifically, researchers across psychology, economics, and sociology suggest that negative intergroup outcomes—between Black and White people most notably—may be rooted, in part, in real and perceived resource competition [8–10]. Surprisingly, despite considerable extant research on group competition, there is a dearth of empirical work examining the causal role of perceived interracial competition on negative psychological outcomes. Indeed, theorists highlight that "*although many studies have documented correlations between such threats* [including intergroup competition] *and intergroup attitudes, experimental and quasi-experimental tests . . . are relatively sparse*" ([11], p. 212). To meet this call,

**Competing interests:** The authors have declared that no competing interests exist.

the present research manipulated perceptions of interracial competition and measured effects on negative psychological outcomes.

## Intergroup competition

Myriad intergroup process models, including realistic group conflict [9], integrated threat [12], construal process [13], and social identity [14] theories, all argue that competition between social groups leads to outgroup threat and negative outcomes. For instance, outgroup threat includes cognitive components, such as zero-sum beliefs, and affective components, such as feelings of anxiety [15]. In response to threats from outgroups, ingroup members exhibit motivation to quell the threats, which can take the form of ingroup favoritism [16], outgroup derogation [17], and behavioral avoidance [18].

Discrimination, or biased treatment of a group or its members [19], functions to promote positive self-regard in agents by either demoting competing outgroups (i.e., outgroup derogation; [14, 20]) or by reserving benefits and favors for ingroups (i.e. ingroup favoritism; [21, 22]. Behavioral avoidance creates physical and/or psychological distance between social groups to reduce the salience of competition [23]. Intergroup competition can also exacerbate preexisting biases and stereotypes that promote mistrust [8, 9]. Taken together, the literature suggests that competition between social groups is associated with myriad negative outcomes.

Not only does *actual* resource competition have negative downstream implications, but *perceived* resource competition can have similar implications for prejudice, stereotyping, and hostility directed towards (perceived) competing outgroups. For example, subjective beliefs about competition and competing outgroups are positively associated with feelings of threat and perceived intergroup biases stemming from ingroup favoritism and outgroup hate [24, 25]. Other research indicates that perceiving outgroup members as competitive is positively associated with intergroup anxiety and negatively associated with prosociality towards outgroup members [26, 27]. Considering the importance of the aforementioned outcomes, as well as their connection with racial disparities (e.g., [28–30]), the work presented here measures perceptions of discrimination, behavioral avoidance, intergroup anxiety, and interracial mistrust as the primary outcomes of interest.

## Interracial competition

Although social groups can be constructed on the basis of multiple factors, race is a particularly salient social group construction in American society [31]. U.S. Census Bureau [32] data predict that non-Hispanic "White" individuals will comprise less than 50% of the U.S. population by 2050, thus interracial competition processes are becoming increasingly relevant for understanding how racial groups orient to and interact with one another. The research presented herein is rooted in the idea that competition between racial groups stems from existing inequalities. For instance, substantial Black-White disparities exist across socioeconomic, educational, vocational, and health domains [33–35]. Importantly, interracial competition processes can exacerbate these disparities [8, 9, 36].

Many associations between general intergroup competition and downstream negative outcomes, such as those reviewed above, can be applied to Black-White interracial competition specifically. For example, perceived interracial competition is associated with lower levels of support for affirmative action programs, higher levels of racial bias and stereotyping, ingroup favoritism, outgroup derogation, and perceived intergroup discord [12, 37–39].[1] Moreover, perceived interracial competition is positively related to intergroup anxiety, conflict, and negative racial attitudes, and Black individuals perceive these outcomes to a greater extent than White individuals [40].

Building on research that has identified associations among perceived intergroup competition processes and negative intergroup outcomes (e.g., [39–41]), we sought to explicate the causal role of perceptions of interracial competition in producing negative intergroup outcomes. We hypothesized that the more Black and White individuals perceive that there is high, relative to low, competition between racial groups in their local environment, the more they will report negative intergroup outcomes (i.e. discrimination, avoidance, anxiety, and mistrust). Supportive data would be important, as a recent review emphasized that empirical work examining the causal role of interracial competition (including perceived interracial competition) on negative psychological outcomes remains sparse [11]. Moreover, understanding the causal factors driving negative intergroup outcomes between Black and White people in America is critical for developing process-focused interventions for improving race relations (e.g., [42]).

## The present research

Two experiments were planned to test hypotheses. Study 1 manipulated perceptions of interracial competition using a normative feedback approach: Participants were informed of group-level perceptions of ongoing interracial competition, ostensibly from members of their community (i.e. ZIP-code). Participants randomly assigned to the high perceptions of competition condition were hypothesized to perceive more discrimination, behavioral avoidance, intergroup anxiety, and interracial mistrust relative to those assigned to the low perceptions of competition condition.

Study 2, which was preregistered, replicated and extended Study 1 by testing whether effects of the group-level competition manipulation manifested in both Black and White participants. We predicted a main effect for race based on previous research [40, 43]: Black participants were expected to report higher levels of each of the four negative interracial outcomes. We also predicted that effects of the manipulation would manifest for both Black and White participants. That is, we theorized that effects of perceptions of interracial competition would not be driven by just one social group, but rather by both competing, mutually involved groups (despite differences in racial attitudes and socioeconomic circumstances; [34, 44, 45]). Finally, given links between competition and income inequality, we also examined the moderating role of local-area racial income gaps. Past work has posited race-based income inequality may make resource and group differences salient, exacerbating perceptions of competition, both inter-individually and between Black and White racial groups [43, 46]. Thus, we tested whether objective, local-area racial income inequality moderated effects of condition on the four focal outcomes (i.e., perceived discrimination, avoidance, anxiety, and mistrust).

Sample sizes were determined *a priori* for both studies. All data were collected before analyses were conducted, and analyses were planned a priori. All manipulations, data exclusions, and variables analyzed are reported for all studies; the data are freely available for download on our lab website (https://socialstresslab.wixsite.com/urochester/research).

## Study 1

Study 1 tested the effect of perceived interracial competition on the four focal negative interracial outcomes: Perceived discrimination, behavioral avoidance, intergroup anxiety, and interracial mistrust.

## Method

**Sample size estimation.** Power analysis revealed that 788 participants (394 per between-subjects condition) were needed to detect a small condition effect ($d = .20$), given a targeted

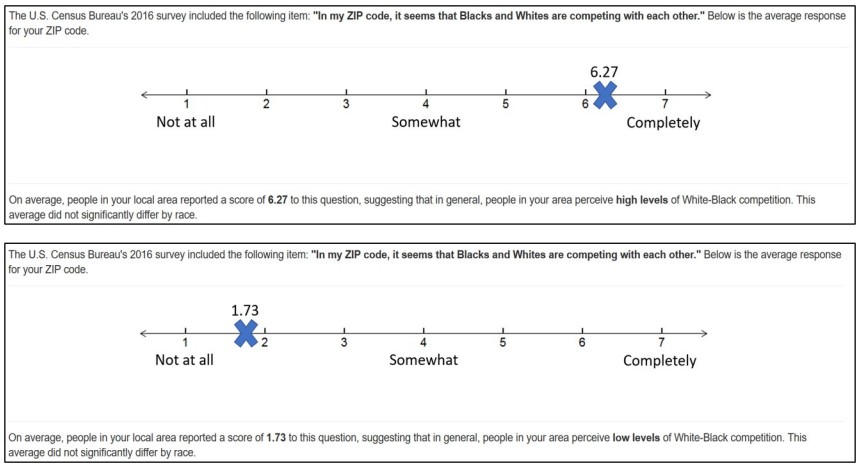

**Fig 1. Display screen for the high and low perceived interracial competition condition based ostensibly on previously entered ZIP-code information.**

power of .80 ($p$ = .05). To account for attention check failures, we sought to oversample by a minimum of 10%.

**Participants.** The recruited sample was 899 U.S. residents. Fifty-two participants failed the attention check and were excluded a priori from analyses, leaving a sample of 847: 451 females, 396 males; 630 White, 67 Black/African-American, 61 Asian, 54 Hispanic, 21 Native American, 2 Pacific Islander, 12 Other; $M_{age}$ = 34.47, $SD_{age}$ = 10.88 (range = 18–86). All data were collected on Amazon's Mechanical Turk; participants were compensated $0.25.[2]

**Procedure.** Procedures were approved by the University of Rochester's Research Subjects Review Board and participants provided consent online prior to participation. A normative feedback approach was used to manipulate perceptions of interracial competition. This approach has been used to manipulate attitudes and perceptions in other areas of research [47–49]. Participants first entered their ZIP-code, which initiated a "calculating" screen for four seconds, followed by (ostensibly) a display of their ZIP-code level census statistics. ZIP-code was used because research suggests that individuals are more accurately aware of sociodemographic information at the local versus state or national level [46, 50]. Moreover, social comparative information is more psychologically meaningful at more local geographic scales [51, 52]. Participants received statistics about their local area, followed by a number line denoting the average self-reported rating of perceived interracial competition for their ZIP-code (anchors ranging from 1 to 7; see Fig 1). After the perceived competition manipulation, participants completed a manipulation check and self-report measures of the negative interracial outcomes. Participants were fully debriefed at the conclusion of the study.

**Measures.** See Table 1 for descriptive statistics and intercorrelations; see S1 Appendix for items.

*Perceived interracial competition (manipulation check).* Murayama and Elliot's [5] five-item perceived competition scale was adapted to fit the race-based focus of the study (e.g. "*In my ZIP code, it seems that Blacks and Whites are competing against each other*"; 1 = *not at all*, 7 = *completely*).

*Perceived discrimination.* The nine-item Everyday Discrimination Scale [53] was adapted. Original instructions read: "*In your day-to-day life how often have any of the following things happened to you because of your race?*"; the adapted scale read: "*In your ZIP code, how often do the following things happen to people because of their race?*". A sample event included: "*Being*

**Table 1. Descriptive statistics and intercorrelations for perceived interracial competition and the race-based psychological outcomes in Study 1.**

| | Descriptive statistics | | | Pairwise intercorrelations | | | | |
|---|---|---|---|---|---|---|---|---|
| | A | M | SD | 1 | 2 | 3 | 4 | 5 |
| 1. Perceived interracial competition | .95 | 3.07 | 1.73 | – | | | | |
| 2. Perceived discrimination | .97 | 3.52 | 1.57 | .55*** | – | | | |
| 3. Perceived behavioral avoidance | .97 | 2.87 | 1.65 | .63*** | .71*** | – | | |
| 4. Perceived intergroup anxiety | .97 | 3.10 | 1.78 | .63*** | .75*** | .86*** | – | |
| 5. Perceived interracial mistrust | .95 | 3.36 | 1.48 | .08* | .35*** | .26*** | .35*** | – |

Notes:

\*\*\*p < .001,

\*p < .05.

*treated with less courtesy than others*" (1 = *never*, 7 = *frequently*). An attention check was included within this scale; specifically, participants read an item asking them to select "2."

*Perceived behavioral avoidance.* Lackey's [54] eleven-item behavioral avoidance scale was adapted (e.g., "*In my ZIP code, Black and White people try to avoid having conversations with each other*"; 1 = *strongly disagree*, 7 = *strongly agree*).

*Perceived intergroup anxiety.* Four items were adapted from Amodio's [55] state affect measure (e.g., "*In my ZIP code, Black and White people feel nervous about interacting with each other*"; 1 = *strongly disagree*, 7 = *strongly agree*).

*Perceived interracial mistrust.* Four items were adapted from the original six-item general trust scale ([56]; e.g., "*In my ZIP code, Black and White people view each other as trustworthy*"; 1 = *not at all*, 7 = *completely*). We reverse-scored responses such that higher values corresponded to higher mistrust; this was done to be consistent with the other negative psychological outcome variables.

## Results

**Manipulation check.** Confirming the effectiveness of the manipulation, participants in the high interracial competition (HRC) condition perceived more interracial competition (M = 3.62) compared to those in the low interracial competition (LRC) condition (M = 2.54), $t(839) = 9.58$, [.86, 1.31] [3], $p < .001$, $d = .66$.

**Effects of perceived interracial competition manipulation on perceived negative interracial outcomes.** Consistent with hypotheses, participants in the HRC condition perceived more discrimination, $t(845) = 5.38$, [.36, .78], $p < .001$, $d = .37$; behavioral avoidance, $t(845) = 4.80$, [.32, .76], $p < .001$, $d = .33$; intergroup anxiety, $t(845) = 5.49$, [.43, .90], $p < .001$, $d = .38$; and interracial mistrust, $t(845) = 3.32$, [.14, .53], $p < .001$, $d = .23$; than those in the LRC condition (see Table 2).

## Discussion

Consistent with hypotheses, perceptions of interracial competition impacted perceptions of each of the four critical interracial outcomes in the expected direction. These data suggest that beliefs that racial groups are competing with one another can directly lead to perceptions of discrimination, avoidance, anxiety, and mistrust. Moreover, this study bolsters existing models of interracial and intergroup competition by providing experimental evidence for downstream ramifications of this kind of competition [7, 12, 38, 39].

Table 2. Means and t-test statistics of perceived interracial competition condition on race-based psychological outcomes in Study 1.

| Variable | Condition High | | Condition Low | | t | df | 95% CI of difference | Cohen's d |
|---|---|---|---|---|---|---|---|---|
| | M | SD | M | SD | | | | |
| Perceived interracial competition | 3.62 | 1.68 | 2.54 | 1.61 | 9.58*** | 839 | [0.86 1.31] | 0.66 |
| Perceived Discrimination | 3.81 | 1.56 | 3.24 | 1.53 | 5.38*** | 845 | [0.36 0.78] | 0.37 |
| Perceived Behavioral Avoidance | 3.15 | 1.64 | 2.61 | 1.62 | 4.80*** | 845 | [0.32 0.76] | 0.33 |
| Perceived Intergroup Anxiety | 3.44 | 1.77 | 2.78 | 1.74 | 5.49*** | 845 | [0.43 0.90] | 0.38 |
| Perceived Interracial Mistrust | 3.53 | 1.45 | 3.19 | 1.49 | 3.32*** | 845 | [0.14 0.53] | 0.23 |

Notes:

***$p < .001$.

## Study 2

Although the Study 1 data supported hypotheses, the sampling method restricted external validity. That is, like most experimental studies of intergroup competition, and interracial competition more specifically, the sample was not equally representative of the competing social groups. Indeed, participants from Study 1 were not screened based on racial/ethnic identity, and only 8% of Study 1's sample identified as Black. Thus, conclusions on whether perceived interracial competition impacts both Black and White individuals similarly could not be made, which is of paramount importance considering the dissimilarities and disparities between these salient racial groups (e.g., [45, 57, 58]). On the one hand, because interracial (and intergroup) competition involves participation and engagement from both ingroups and outgroups, Black and White people may be similarly influenced by perceptions of interracial competition. On the other hand, competition effects may differ as a function of racial group membership. That is, because Black people experience worse outcomes compared to White people across numerous social, psychological, and economic indicators, it is possible that perceived interracial competition disproportionally impacts Black individuals. Alternatively, because high-status groups (e.g., White people) have more to lose from shifting status relations [59], White people may be more likely to be affected by rising perceptions of interracial competition. Past research, however, suggests that White individuals feel less competitive threat from Black individuals compared to other racial groups [60]. Nonetheless, to address lingering questions tied to racial groups' responses to competition, Study 2 recruited similar numbers of Black and White participants and examined whether effects of interracial competition differed across racial groups.

One notable dimension in which Black and White people exhibit a large disparity is in income: Black individuals earn substantially less than White individuals [57]. Income gaps between groups can make resource differences more salient, promoting competition [7, 24, 25]. Broadly, income inequality describes one's relative economic position compared to relevant others. Grounded in social comparison processes [61–63], individuals leverage information about relevant others to gauge their position in status hierarchies. Under extreme cases of economic disparities, relative comparisons on income become particularly salient [51], and have the potential to discourage reciprocity, reinforce consumption norms, and increase perceptions that individuals and social groups (e.g., racial groups) are competing with one another [43, 46, 64, 65].

Like generalized income inequality, racial income inequality (e.g., Black-White income inequality) can also influence perceptions of competition. Importantly, as social comparison processes can occur between social groups [66, 67], and given the prominence of race and

Black-White relations in the United States, race-based comparisons are likely to occur in the context of limited resources (e.g., money). The combination of perceived limited resources, which are exacerbated by the prevalence of Black-White income gaps, and the presence of a relevant, comparative outgroup are natural precursors of intergroup competition [23]. Moreover, these effects emerge for both competing groups. Disadvantaged individuals feel deprived of important outcomes [68–71], and advantaged individuals are concerned about losing social position and seek to maintain it [72–74]. Supporting this view, past research has documented associations between the Black-White income gap and perceived interracial competition, which held for both White and Black individuals [43]. If race-based income inequality has the potential to enhance competition by making group differences and resource differentials salient, outcomes that stem from competition, including but not limited to perceived discrimination, avoidance, anxiety, and mistrust, may be exacerbated. Towards this end, Study 2, which was preregistered (https://aspredicted.org/ay8dc.pdf), also examined the moderating role of racial income inequality on condition effects.

## Method

**Sample size estimation.** An a priori power analysis revealed that 1,576 participants (394 per between-subjects condition, per racial group) were needed to detect a small condition effect ($d$ = .20) given a targeted power of .80 ($p$ = .05). To account for attention check failures, we oversampled by at least 10%.

**Participants.** The recruited sample was 1,823 U.S. residents. One hundred seventy-eight participants failed to complete the attention check, improperly completed the demographic questionnaire, or completed the survey more than once. These participants were excluded a priori from analyses, leaving a sample of 1,645 (975 females, 669 males; 836 White, 809 Black/African-American; $M_{age}$ = 36.46, $SD_{age}$ = 11.75 (range = 18–78). As in Study 1, all data were collected on Amazon's Mechanical Turk and participants were compensated $0.25.

**Procedure and measures.** Procedures and measures were identical to those reported in Study 1.

*Racial income gap (RIGap).* Because participants entered their ZIP-code in order to receive the normative information induction, we were able to use their ZIP-codes to calculate the degree of racial income inequality in their area. The RIGap was calculated using the 2016 American Community Survey's five-year estimates (the most recent estimates available during data collection). These data are publicly available from the U.S. Census Bureau (https://data.census.gov/cedsci/). A gap score was calculated using the income difference between Black and White people in a given ZIP-code area. Higher values correspond to White individuals having more income on average compared to Black individuals.

## Results

**Manipulation check.** A 2 (Condition) x 2 (Race) analysis of variance (ANOVA) was used to test effects on the dependent measures (see Table 3 for means and standard deviations). Consistent with Study 1, participants in the HRC condition perceived more interracial competition ($M$ = 3.65) compared to those in the LRC condition ($M$ = 2.44), $F(1, 1641)$ = 242.13, $p <$ .001, $\eta p^2$ = .13. Moreover, race was significant, $F(1, 1641)$ = 25.62, $p <$ .001, $\eta p^2$ = .015; Black participants reported greater perceptions of interracial competition than White participants ($Ms$ = 3.25 vs. 2.87).

**Effects of perceived interracial competition manipulation on negative psychological outcomes.** See Table 4 for a results summary. As predicted, and consistent with Study 1, relative to those assigned to the LRC condition, participants assigned to the HRC condition

**Table 3. Descriptive statistics and intercorrelations for perceived interracial competition, the race-based psychological outcomes, and racial income gap in Study 2.**

| | Descriptive statistics | | | Pairwise intercorrelations | | | | | |
| --- | --- | --- | --- | --- | --- | --- | --- | --- | --- |
| | α | M | SD | 1 | 2 | 3 | 4 | 5 | 6 |
| 1. Racial income Gap | – | $10,340 | $14,902 | – | | | | | |
| 2. Perceived interracial competition | .95 | 3.06 | 1.68 | .04 | – | | | | |
| 3. Perceived discrimination | .96 | 3.52 | 1.53 | .03 | .53*** | – | | | |
| 4. Perceived behavioral avoidance | .97 | 2.65 | 1.50 | .04 | .55*** | .63*** | – | | |
| 5. Perceived intergroup anxiety | .97 | 2.93 | 1.66 | .05 | .55*** | .68*** | .82*** | – | |
| 6. Perceived interracial mistrust | .95 | 3.67 | 1.48 | -.04 | .17*** | .42*** | .29*** | .38*** | – |

Notes:

***$p < .001$,

**$p < .01$, *$p < .05$.

perceived more discrimination, $F(1, 1641) = 71.63$, $p < .001$, $\eta p^2 = .042$, behavioral avoidance, $F(1, 1641) = 38.84$, $p < .001$, $\eta p^2 = .023$, intergroup anxiety, $F(1, 1641) = 69.85$, $p < .001$, $\eta p^2 = .041$, and interracial mistrust, $F(1, 1641) = 21.89$, $p < .001$, $\eta p^2 = .013$. Race was also significant; Black participants perceived more discrimination, $F(1, 1641) = 62.57$, $p < .001$, $\eta p^2 = .037$, behavioral avoidance, $F(1, 1641) = 4.08$, $p = .044$, $\eta p^2 = .002$, intergroup anxiety, $F(1, 1641) = 10.14$, $p = .001$, $\eta p^2 = .006$, and interracial mistrust, $F(1, 1641) = 123.77$, $p < .001$, $\eta p^2 = .070$. No Condition x Race interactions emerged for any of the outcome variables ($Fs < .43$, $p$s $> .51$). [4]

**Table 4. Means and ANOVA analysis of perceived interracial competition condition and race on race-based psychological outcomes in Study 2.**

| | | Condition High | | Condition Low | | | Mean Square | F | $\eta p^2$ |
| --- | --- | --- | --- | --- | --- | --- | --- | --- | --- |
| | Race | M | SD | M | SD | Effect | | | |
| Perceived interracial competition | White | 3.39 | 1.69 | 2.32 | 1.48 | Condition | 597.76 | 232.62*** | .124 |
| | Black | 3.92 | 1.61 | 2.57 | 1.47 | Race | 63.25 | 25.62*** | .015 |
| | | | | | | Interaction | 7.73 | 3.13† | .002 |
| Perceived Discrimination | White | 3.51 | 1.45 | 2.95 | 1.36 | Condition | 154.65 | 71.63*** | .042 |
| | Black | 4.13 | 1.48 | 3.47 | 1.58 | Race | 135.07 | 62.57*** | .037 |
| | | | | | | Interaction | 0.92 | 0.43 | .000 |
| Perceived Behavioral Avoidance | White | 2.80 | 1.55 | 2.34 | 1.46 | Condition | 85.87 | 38.84*** | .023 |
| | Black | 2.95 | 1.46 | 2.49 | 1.47 | Race | 9.02 | 4.08* | .002 |
| | | | | | | Interaction | 0.01 | 0.05 | .000 |
| Perceived Intergroup Anxiety | White | 3.14 | 1.67 | 2.47 | 1.53 | Condition | 184.18 | 69.85*** | .041 |
| | Black | 3.39 | 1.64 | 2.72 | 1.64 | Race | 26.73 | 10.14** | .006 |
| | | | | | | Interaction | 0.01 | 0.00 | .000 |
| Perceived Interracial Mistrust | White | 3.46 | 1.45 | 3.10 | 1.41 | Condition | 44.33 | 21.90*** | .013 |
| | Black | 4.21 | 1.33 | 3.92 | 1.50 | Race | 250.59 | 123.77*** | .070 |
| | | | | | | Interaction | 0.38 | 0.19 | .000 |

Notes:

***$p < .001$,

**$p < .01$,

*$p < .05$,

†$p < .10$.

**Table 5. Standardized coefficient estimates of the condition and race on race-based outcomes moderated by racial income gap in Study 2.**

| | PCOMP | | | | DISCRIM | | | | AVOID | | | | ANX | | | | MISTRUST | | | |
|---|---|---|---|---|---|---|---|---|---|---|---|---|---|---|---|---|---|---|---|---|
| | Step 1 | | Step 2 | | Step 1 | | Step 2 | | Step 1 | | Step 2 | | Step 1 | | Step 2 | | Step 1 | | Step 2 | |
| Variable | β | SE | β | SE | β | SE | B | SE | β | SE | β | SE | β | SE | β | SE | β | SE | β | SE |
| Condition | .37*** | .02 | .36*** | .02 | .20*** | .02 | .20*** | .02 | .16*** | .02 | .15*** | .02 | .21*** | .02 | .21*** | .02 | .11*** | .02 | .12*** | .02 |
| Race | -.11*** | .02 | -.10*** | .02 | -.19*** | .02 | -.19*** | .02 | -.04 | .02 | -.04 | .02 | -.07** | .02 | -.07** | .02 | -.26*** | .02 | -.26*** | .02 |
| RGAP | .03 | .02 | .04 | .02 | .02 | .02 | .03 | .03 | .03 | .02 | .04 | .03 | .04 | .02 | .05 | .02 | -.04 | .02 | -.04 | .02 |
| Condition x RGAP | | | .07** | .02 | | | .04 | .02 | | | .06* | .02 | | | .05* | .02 | | | .00 | .02 |
| Race x RGAP | | | -.03 | .02 | | | -.03 | .03 | | | -.04 | .03 | | | -.04 | .02 | | | .04 | .02 |

Notes:

***$p < .001$,

**$p < .01$,

*$p < .05$.

**Bayes factor—Model comparisons.** Because we were interested in whether the main effects model was preferred to the interaction model (i.e. support for the absence of an interaction effect), we calculated Bayes factors (BF10) to estimate the comparative strength of each model. These Bayes factors allow one to assess the evidence against the inclusion of an interaction term [75]. Bayes factors were calculated with using JASP software [76]. Data demonstrated strong evidence against including the interaction term for each of the four focal outcomes by roughly a factor of 10 (discrimination = 10.62, behavioral avoidance = 12.80, anxiety = 12.28, mistrust = 10.09; [75]). Thus, the data are more likely under the main effects than interaction model, meaning that it can be concluded that Black and White participants exhibited similar condition effects on psychological outcomes.

**Moderation of racial income gap.** Hierarchical multiple regression analyses were used to examine the effects of condition, race, and racial income gap on each of the outcome variables separately. Condition, race, and RIGap was entered in step 1, followed by Condition x RIGap and Race x RIGap interaction terms in step 2. For parsimony, we focus herein on the Condition x RIGap interactions (see Tables 5 and 6).

Given the hierarchical structure of the data (participants nested in ZIP-codes), we first built a multilevel model having no predictor, using each of the outcomes separately. We first calculated the design effect (DEFF; [77]); this assessed the impact of ZIP-code clustering on estimation of the standard error. A DEFF > 2 indicates that the impact of ZIP-code clustering is

**Table 6. Standardized coefficient estimates of the simple slopes for the condition x racial income gap interaction on race-based psychological outcomes in Study 2.**

| | Racial Income Gap (+1 SD) | | Racial Income Gap (-1 SD) | |
|---|---|---|---|---|
| Outcome | B | CI | β | CI |
| Perceived interracial competition | .44*** | [.37, .50] | .30*** | [.23, .36] |
| *Perceived Discrimination* | .25*** | [.18, .31] | .16*** | [.09, .22] |
| Perceived Behavioral Avoidance | .22*** | [.15, .29] | .09** | [.02, .16] |
| Perceived Intergroup Anxiety | .26*** | [.19, .33] | .16*** | [.09, .22] |

Notes:

***$p < .001$,

**$p < .01$;

italicized outcomes qualified by marginally significant interactions.

substantial and that multilevel analyses should be preferred over single-level analyses [78]. Across each of the outcomes, the design effects ranged from 1.02–1.05. As the design effect was well below threshold, the incidence of ZIP-code clustering was negligible, indicating that single-level and multilevel analyses were expected to produce comparable results. Thus, we used single-level analyses.

Condition x RIGap predicted perceptions of interracial competition, $\beta = .07$, [.02, .11], $p = .003$, behavioral avoidance, $\beta = .06$, [.02, .11], $p = .01$, and intergroup anxiety, $\beta = .05$, [.003, .10], $p = .038$. The effects of condition on these outcomes were stronger for those living in areas with larger racial income inequality.

## Discussion

Consistent with hypotheses and Study 1, high levels of perceived interracial competition led to more perceptions of discrimination, behavioral avoidance, intergroup anxiety, and interracial mistrust compared to low levels of perceived interracial competition. Moreover, effects manifested similarly for both Black and White participants, suggesting that the causal effects of perceived interracial competition on psychological outcomes may be independent of relative social group position (advantaged vs. disadvantaged) and/or group processes, such as stigmatization or health disparities (to name a few). In addition, main effects of race also emerged; Black participants reported higher levels of all negative psychological outcomes, replicating past findings [40, 43].

Racial income inequality moderated a subset of effects. Those living in areas with greater objective racial income inequality were more strongly impacted by the manipulation, exhibiting greater perceived avoidance and intergroup anxiety. These results suggest that individuals perceive (either implicitly or explicitly) Black-White income inequality, which influences how perceptions of interracial competition impact negative psychological outcomes.

## General discussion

This research tested the effects of manipulating group-level perceptions of interracial competition on interracial psychological outcomes. Study 1 found that individuals assigned to receive information indicating that interracial competition is high in their community perceived more discrimination, behavioral avoidance, intergroup anxiety, and interracial mistrust. Study 2 replicated the findings from Study 1 in Black and White samples, and also tested racial income inequality as a moderator, which demonstrated that the manipulation had a stronger effect on individuals living in communities with greater levels of interracial income inequality. Building on existing correlational research on intergroup competition [38–40], these data support the notion that perceptions of interracial competition can operate as a causal antecedent of negative interracial outcomes.

This research also highlighted the ubiquity of interracial competition effects. That is, manipulated perceptions of interracial competition impacted Black and White individuals similarly. Although Black individuals experience worse outcomes compared to White individuals across numerous domains (e.g. education, job attainment, and healthcare) and perceive more negative psychological outcomes overall (see Study 2), members from both racial groups perceive negative outcomes as a function of increased perceptions of competition. The similar effects of the competition manipulation on Black and White participants is important because the negative psychological variables tested here have the potential to maintain, or exacerbate, societal-level disparities, and engender further competition between groups. Furthermore, our results indicate that perceived interracial competition is particularly pernicious in highly unequal contexts. When the income distribution between Black and White people is greater,

competition may be experienced as both a producer and sustainer of group disparities, with strong negative downstream implications for intergroup affect, cognition, and behavior.

Broadly, the findings presented here contribute to the literature on competition, race, and negative psychological outcomes. Notably, the vast majority of research to date on these topics has been correlational [11], thus the experimental effects observed here confirm many underlying assumptions in models of intergroup competition. Moreover, understanding how perceptions of interracial competition causally elicit negative intergroup perceptions can help inform the development of process-focused interventions for improving race relations (e.g., [42]) and, more downstream, attenuating group disparities between Black and White people.

## Implications for theory development

Past research has documented effects of competition on approach and avoidance motivation [5, 79]. Approach motivation entails the energization or direction of behavior toward desirable objects, situations, or outcomes, while avoidance motivation entails the energization or direction of behavior away from undesirable objects, situations, or outcomes [80, 81]. Along these lines, strategies to cope with the experience of competition can vary along approach/avoidance dimensions in interracial competitions. Specifically, individuals can engage both approach and avoidance action tendencies to influence social position in response to competition [82–84]. In interracial competitions these tendencies could manifest as approach-oriented affective responses such as discrimination, anger, and risk taking [14, 39, 85], or avoidance-oriented responses such as intergroup anxiety and outgroup avoidance [18, 86]. Additionally, understanding how perceptions of interracial competition shape motivational processes may have important implications for health (e.g., [85, 87, 88]). Although the present findings suggest that perceptions of interracial competition can elicit approach- and avoidance-oriented responses, additional work is needed to integrate intergroup/interracial action tendencies and these approach/avoidance motivational processes.

Each of the psychological outcome variables examined here may also inform research on health and racial disparities [28–30]. For example, Black individuals who perceive more discrimination are more likely to engage in substance use [28], and perceptions of being discriminated against predicts worse health outcomes [89, 90]. Avoidance behavior is reflected in residential segregation [91, 92], which may be stronger for White people avoiding Black people [93]. Subsequently, residential segregation is linked to other negative outcomes, such as worse educational and health outcomes for Black individuals relative to White individuals [94, 95]. Intergroup anxiety—an affective process—has myriad negative psychological, behavioral, and health consequences [96–98]. Notably, anxiety impairs performance, shifts attention to negative cues, and predicts poor biological functioning [99–101]. Lastly, Black individuals perceive more mistrust than White individuals, which engenders negative evaluations of White people [102, 103]. Although little is known about the direct relation between intergroup mistrust and health, there is a negative association between perceptions of mistrust and the experience of threat, which can negatively impact health (e.g., [104]).

## Limitations and future directions

Limitations should be considered when interpreting our findings. First, our work focuses on *perceived* outcomes, which does not allow for definitive conclusions regarding more downstream *objective* intergroup outcomes, such as behavioral or biological outcomes, or disease prevalence. As such, future research would do well to link perceived intergroup competition to more objective, societal-level outcomes, such as drug use and violent crime rates [105–107]. Importantly, this work would need to focus on measured perceptions of competition, as it is

unlikely that a temporarily manipulated perception of competition would have real-world objective implications, unless manipulated perceptions are internalized over time (e.g., [108]).

Second, the interracial outcomes measured here reflect perceptions of the prevalence in one's community. That is, participants did not report on how much they discriminate or avoid outgroups, but rather on the prevalence of these in their community. This approach mirrors much work in the stereotyping, prejudice, and discrimination literatures, which implements similar methods to avoid socially desirable responding (e.g., [109–111]). However, some recent research suggests that probing with targeted, direct questions pertaining to prejudicial, stereo-typical, and discriminatory outcomes may, in fact, be highly informative [112–114]. Thus, future research may benefit from examining more person-level consequences of perceived interracial competition on being the agent and target of intergroup anxiety, avoidance, mistrust, and discrimination.

Third, the present research directly manipulated perceptions of interracial competition. Thus, a possible limitation of this approach is that metacognitive rather than intra-cognitive processes may be driving effects. However, this approach was intentional, as the primary goal was to examine the effect of *perceived* interracial competition on *perceived* interracial outcomes. A method such as having participants read articles about Black-White discord may seem a viable option for our perception manipulation, but such an approach would seek to induce perceptions of interracial competition by manipulating outcomes of competition, which would deviate from our proposed causal sequence.

Fourth, to manipulate perceived interracial competition, a brief normative feedback approach was used. Most studies in the intergroup competition literature, however, focus on realistic or symbolic threat inductions to activate perceptions of intergroup competition and conflict (for a review, see [11]). That is, conflict and competition are often intertwined. The manipulation used herein, however, provided no information about conflict, only competition. Even so, this minimalist approach was sufficient to change perceptions of intergroup psychological outcomes, but we caution against overgeneralizing findings to group conflict contexts. Moreover, our experimental paradigm was agnostic with regard to the locus of the interracial effect—that is, one group was not presented as competing with another group, rather the groups were simply presented as competing *with* each other. Similar to the threat-based paradigms highlighted above, understanding unidirectional processes is potentially important because the actions and experiences of Black and White people often diverge [40, 93].

Finally, this work focused exclusively on Black-White relations, and it would be informative to extend the work to other majority-minority relationships, such as White and non-White Hispanic groups. Supporting this avenue, research shows that the growth of the Latinx population is a significant predictor of feelings of threat among White Americans [115]. Such research would allow testing of the generalizability and nuances of the intergroup effects observed herein.

## Conclusion

This research documented the causal role of perceived interracial competition on interracial psychological outcomes. This work contributes to our current understanding of group competition in that it identifies perceived interracial competition as a causal antecedent of perceived interracial discrimination, avoidance, anxiety, and mistrust. It is paramount that work in this area continues to elucidate how macro- and group-level psychological processes influence individuals at the person-level, especially in such a critically important area of research—that of Black-White relations in America.

## Supporting information

**S1 Appendix. This appendix contains scales and measures used in Studies 1 and 2.**
(DOCX)

**S1 File. This file contains footnotes and ancillary analyses looking at moderation effects of a subset of relevant variables.**
(DOCX)

**S2 File.**
(PDF)

**S1 Data.**
(XLSX)

**S2 Data.**
(SPS)

**S3 Data.**
(SPS)

**S4 Data.**
(XLSX)

**S5 Data.**
(SPS)

## Author Contributions

**Conceptualization:** Jonathan Gordils, Andrew J. Elliot, Jeremy P. Jamieson.

**Data curation:** Jonathan Gordils.

**Formal analysis:** Jonathan Gordils.

**Methodology:** Jonathan Gordils, Andrew J. Elliot, Jeremy P. Jamieson.

**Project administration:** Jonathan Gordils.

**Supervision:** Andrew J. Elliot, Jeremy P. Jamieson.

**Writing – original draft:** Jonathan Gordils, Andrew J. Elliot, Jeremy P. Jamieson.

**Writing – review & editing:** Jonathan Gordils, Andrew J. Elliot, Jeremy P. Jamieson.

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
