## [Decision Letter · Decision Letter 0]

23 Sep 2020

PONE-D-20-22299

The effect of perceived interracial competition on psychological outcomes

PLOS ONE

Dear Dr. Gordils,

Thank you for submitting your manuscript to PLOS ONE. After careful consideration, we feel that it has merit but does not fully meet PLOS ONE’s publication criteria as it currently stands. Therefore, we invite you to submit a revised version of the manuscript that addresses the points raised during the review process.

In particular, both reviewers raised concerns about the conceptualization of the dependent variables and conclusions/implications of these findings. As participants' intergroup anxiety, mistrust, behavioral avoidance, or discrimination tendency were not assessed, the discussion should be more tentative in asserting that these findings have implications for the effect of perceived competition on individuals' intergroup behavior. Also, please specify how Study 2 participants were recruited (MTurk?) and whether MTurk data were screened for issues that have been documented with this platform (e.g., duplicate IP addresses or geolocations).   

We look forward to receiving your revised manuscript.

Kind regards,

Natalie J. Shook

Academic Editor

PLOS ONE

Journal Requirements:

2.Thank you for including your ethics statement:  "University of Rochester Research Subjects Review Board (UR RSRB)

No. STUDY00001771

Consent was provided electronically and all data were analyzed anonymously.".   

1. Please amend your current ethics statement to confirm that your named institutional review board or ethics committee specifically approved this study.

3. We noted in your submission details that a portion of your manuscript may have been presented or published elsewhere.

[Yes.

A subset of Study 1 Data is being used in a larger compiled dataset (N = 2333) to examine pre vs post COVID-19 effects, population density, and political orientation on negative interracial outcomes in a separate manuscript under review in a different journal. The focal interests of this other manuscript are completely separate of this current manuscript, although we believe it is important to be transparent on this issue.]

Reviewers' comments:

Reviewer's Responses to Questions

**Comments to the Author**

1. Is the manuscript technically sound, and do the data support the conclusions?

Reviewer #1: Yes

Reviewer #2: Partly

2. Has the statistical analysis been performed appropriately and rigorously? 

Reviewer #1: Yes

Reviewer #2: Yes

3. Have the authors made all data underlying the findings in their manuscript fully available?

Reviewer #1: Yes

Reviewer #2: Yes

4. Is the manuscript presented in an intelligible fashion and written in standard English?

Reviewer #1: Yes

Reviewer #2: Yes

5. Review Comments to the Author

Reviewer #1: Review PONE-D-20-22299

The effect of perceived interracial competition on psychological outcomes

In this paper the authors answered a call by the field to investigate the causal nature of intergroup (specifically interracial) competition on negative psychological outcomes, specifically perceived discrimination, perceived behavioral avoidance, perceived intergroup anxiety, and perceived interracial mistrust. In Study 1 they recruited people of all racial groups, randomly assigned them to read about the Black-White perceived interracial competition in their local area as either being high or low, and filled out a series of measures tapping into negative psychological outcomes. In a pre-registered Study 2 they specifically recruited only Black and White participants and repeated the interracial competition manipulation.

It’s rare to read a paper that claims that there is a dearth of research on a fairly obvious topic and upon reflection realize there actually isn’t much work that specifically centers competition in intergroup relations! I do believe there is a fair amount of work using minimal group paradigms that manipulates competition, but that work focuses on behavioral or attitudinal outcomes, not necessarily these cognitive mechanisms discussed here.

Along many dimensions I enjoyed reading this manuscript. There were some places where things could be clearer (and I outline them below) but on the whole I think this is an important topic of inquiry. My main issue with this paper (and one that can likely be addressed by a lengthier discussion or with another study) is that I felt like the movement on their four psychological variables can be understood as additional manipulation checks rather than showing a causal pathway from perceived competition to perceived discrimination, behavioral avoidance, intergroup anxiety, and interracial mistrust. To their credit, the authors discuss this in the general discussion section (point four) but I wasn’t completely convinced that there was a large difference between their manipulation check and their four main DVs. The correlations for all but the mistrust items were rather high as well. In terms of what this alternative understanding could have looked like from a participant’s perspective; if you tell me that two groups are competing in my neighborhood, it makes perfect sense that these groups also discriminate against one another, avoid one another, have anxiety about one another, and mistrust one another. I would believe that purely based on my assumptions about how competition works, not at all because perceiving such competition has led me and my beliefs about racial relations to shift.

This alternative hypothesis is easier to generate because the manipulation and the DVs are so similar. The authors told participants that the average response to the question “Blacks and Whites are competing with each other” was either “completely” or “not at all” (i.e. telling participants that Black and White people have terrible or great relations with one another) and then effectively measured perceived relations between the two groups. Here is where not asking about the participants’ beliefs is a hindrance to testing the actual hypothesis. The “perceived” wording/third-party nature of the questions makes social desirability a concern. This becomes more important for Study 1 because there are various racial groups present but the study focuses specifically on the Black-White divide. There isn’t any real discussion around what it means for an Asian or a Hispanic person to answer these questions. The fact that condition (and not race) is the primary driver of the effects is another reason why I feel the alternative hypothesis – that participants are not actually changing their beliefs but reporting beliefs in line with the information told to them by the researchers – might have some merit. The work that does exist on how White people respond to threat (and experiencing competition is usually threatening) is varied. Sometimes they aggressively respond (think shifting demographics and status threat) while other times they move towards appeasement. The equal movement for anxiety, discrimination, mistrust, and avoidance for Black and White people was very surprising from that perspective.

I am a bit wordier than I usually am in reviews regarding this point because I’m trying to be as clear as possible. A lengthier discussion regarding how we should interpret the findings would go a long way to deal with my concern, or to run another study in which they manipulate beliefs about competition and assess participants’ actual discrimination and avoidant behaviors and/or feelings of mistrust and anxiety.

Other (small) points:

1. I had a hard time knowing who the participants’ were based on the abstract and the introduction. The authors wrote that they extended Study 1 in Study 2 by recruiting Black and White participants, which makes an assumption that Study 1 had only Black or White subjects. That’s actually not true so more clarity in writing about the population pool would be helpful.

2. The authors discussed a variety of intergroup process models but don’t include social dominance theory although it explicitly centers competition. I make this point because the authors do include several SDT-centered studies as evidence for their claims.

3. The authors discuss negative intergroup outcomes in a general sense throughout the introduction, but it was difficult to understand if the authors believed that there are negative outcomes for all groups in conflict, differential ones for low power versus high power groups, etc. This is part of why I thought Study 1 had only White subjects because some of these findings are specific to White participants. For example, the authors wrote “perceived interracial competition is associated with lower levels of support for affirmative action programs, higher levels of racial bias and stereotyping, ingroup favoritism, outgroup derogation, and perceived intergroup discord”. However, I assume that Black people don’t reduce support for affirmative action programs when they perceive interracial competition.

4. I loved the inclusion of Bayes factor calculations!

Reviewed by Sa-kiera T. J. Hudson, PhD

Postdoctoral Fellow at Yale University

Sa-kiera.hudson@yale.edu

Blame any and all typos on COVID-19

Reviewer #2: Review of PONE-D-20-22299

This manuscript presents the results of an online study of the impact of manipulated information concerning normative perceptions of interracial competition on participants’ perceptions of elements of discrimination. Study participants were told that a census survey had indicated that people who lived in their zip code thought there either was or was not a lot of Black – White competitiveness in that area. The participants responded by saying there was more or less discrimination, respectively, as well as other manifestations of racism (avoidance of the other group, intergroup “anxiety,” and interracial mistrust). The authors conclude that their results “..documented the causal role of perceived interracial competition on interracial outcomes.”

The study has several strengths. The issue is an important one, and the study is theory-based. The sample size for the first study was disproportionately White (a Black : White ratio of > 9 : 1), but the size for the second study is definitely adequate—being based on an a priori power analysis. Moreover, the second study was preregistered, which is a plus.

However, there are problems with some of the logic behind the hypotheses and the discussion of the relevant theories. For example, it is not clear (from the Introduction) exactly why inequality leads to competition; more specifically, why are Whites competitive with Blacks (is it because they want to increase the advantage that they have; is it guilt-driven in an effort to justify the inequities)? Usually one thinks of competition being heightened by equality—e.g., two good teams competing for first place. If the discrimination is overwhelmingly directed against Blacks, wouldn’t they be likely to be more competitive? On the other hand, competitiveness leading to more outgroup discrimination—for both Blacks and Whites-- is definitely intuitive (as a number of previous studies have shown).

In terms of measures, there is some ambiguity as well. The perceived discrimination measure asked how often people in your zip code had these discriminatory experiences. That is a very different question than one asking how much discrimination the respondent has experienced themselves. One would assume that more egalitarian people would be inclined to: see more discrimination and more inequity in their neighborhoods, and report having experienced less discrimination themselves. A suggestion here: the authors should control for the percentages of the different racial / ethnic groups and also for SES levels in the different zip codes.

A more concerning issue is that the authors are claiming that they manipulated perceived competition between Blacks and Whites, but the manipulation was heavy-handed enough (mean reported agreement with the statement “In my zip code, it seems that blacks and whites are competing with each other” of 6.27 vs. 1.73 on the 1 – 7 scale—a very large difference) that respondents had to assume that there was more and less discrimination in their neighborhoods. The correlations between perceived competition and the various measures of perceived discrimination attest to this. The latter was, in some sense, almost a logical inference from the information the participants had been provided. Compared with this manipulation (ie, its strength), the respondent reports of discrimination and even of interracial competition are not nearly as pronounced. Also, given the strength of the correlations among the different measures of discrimination, the authors might want to combine them into an index.

A couple of very minor points:

- Competition has a number of different connotations (some of which come close to those of the- word conflict).

- I would suggest taking out the assessment of the study on p. 23: “..the experimental effects observed here represent a welcome addition to the literature…”

- same for the use of the term “causal.”

Overall, this is an interesting study on an important topic. In several respects, the authors followed good methodological procedures. However, there are some issues with the wording of the IVs vis a vis the DVs that present some problems.

6. PLOS authors have the option to publish the peer review history of their article (what does this mean?). If published, this will include your full peer review and any attached files.

Reviewer #1: **Yes: **Sa-kiera T J Hudson

Reviewer #2: No

---

## [Author Response · Author response to Decision Letter 0]

27 Oct 2020

Please see attached "Response to Reviewers" document. Additionally, we pasted the text from this document below.

Natalie J. Shook, PhD 

Academic Editor - PLOS ONE

Dear Dr. Shook, 

Thank you for the opportunity to revise our manuscript, “The effect of perceived interracial competition on psychological outcomes" (PONE-D-20-22299). We are pleased that you and the reviewers appreciate this work and are grateful for the feedback received. We have revised the manuscript in response to the thoughtful comments, and we believe the revision represents an improvement over the original submission. Below, we address each of your and the reviewers’ comments in turn (we have copied and pasted the comments below and provide our response to each) and indicate where changes to the manuscript were made. 

Editor Comments

1. In particular, both reviewers raised concerns about the conceptualization of the dependent variables and conclusions/implications of these findings. As participants' intergroup anxiety, mistrust, behavioral avoidance, or discrimination tendency were not assessed, the discussion should be more tentative in asserting that these findings have implications for the effect of perceived competition on individuals' intergroup behavior.

We appreciate this comment. While we believe that examining objective person-level outcomes would be a valued future scientific inquiry, the perceived outcomes reported here have value for elucidating how racial income inequality impacts psychological processes. The revised manuscript seeks to clarify the focus on perceived outcomes. In particular, we include two paragraphs prominently at the outset of the limitations section that focus on 1) perceived vs. objective outcomes and 2) general perceptions vs. self perceptions. 

2. Also, please specify how Study 2 participants were recruited (MTurk) and whether MTurk data were screened for issues that have been documented with this platform (e.g., duplicate IP addresses or geolocations).

To address these concerns, we added information pertaining to duplicate Amazon Turk IDs and specify that data was collected via MTurk. In Study 1, we did not have any duplicate IDs, so this was not a concern. Regarding duplicate IP addresses, while there is debate regarding this potential issue (Moss & Litman, 2018; for an example retaining duplicate IP addresses, see Casey et al., 2017), one major concern with filtering duplicate IP addresses is the potential for the introduction of bias. Lower SES households are more apt to share internet-accessible devices, and thus we would disproportionately exclude those individuals from the research with an IP filtering method.

Regarding geocoding on MTurk, we also agree that this has the potential of being a concern (e.g., Kennedy et al., 2018). However, given the moderation of RIGap in Study 2, and past work using similar methods and finding similar associations (Gordils et al., 2020), we are confident that the work presented here is likely to represent actual participants reporting their actual ZIP-codes. However, if you feel strongly that participants should be screened and filtered by IP address, we are willing to include those tests alongside the nonfiltered results. 

Reviewer #1:

1. My main issue with this paper (and one that can likely be addressed by a lengthier discussion or with another study) is that I felt like the movement on their four psychological variables can be understood as additional manipulation checks rather than showing a causal pathway from perceived competition to perceived discrimination, behavioral avoidance, intergroup anxiety, and interracial mistrust. To their credit, the authors discuss this in the general discussion section (point four) but I wasn’t completely convinced that there was a large difference between their manipulation check and their four main DVs. The correlations for all but the mistrust items were rather high as well. In terms of what this alternative understanding could have looked like from a participant’s perspective; if you tell me that two groups are competing in my neighborhood, it makes perfect sense that these groups also discriminate against one another, avoid one another, have anxiety about one another, and mistrust one another. I would believe that purely based on my assumptions about how competition works, not at all because perceiving such competition has led me and my beliefs about racial relations to shift.

We agree that the outcomes measured here are necessarily related, as past literature would suggest (Sherif, 1966; Stephan & Stephan, 2000; Esses et al., 2005). We also agree with your reasoning. That is, your assumption that “I would believe that purely based on my assumptions about how competition works, not at all because perceiving such competition has led me and my beliefs about racial relations to shift” is consistent with our model. If those negative psychological outcomes are the “baggage” of intergroup competition, that suggests perceptions of intergroup competition are causal processes in eliciting changes in those processes. The experimental paradigm used here directly shows that perceptions of intergroup competition can be manipulated at the local level, and this manipulation has a direct impact on psychological outcomes. That is, in addition to including both White and Black participants and examining important moderation, the novelty of this work stems from the experimental approach, which (to our knowledge) is the first to support the (perhaps intuitive) finding that perceptions of interracial competition directly affect perceptions of intergroup outcomes. As you stated, prior work on group processes and intergroup relations have thus far just assumed these effects. 

Regarding the notion of competition necessitating discrimination, avoidance, and so forth, it is also important to note that competition isn’t inherently negative, but rather provides a context in which these kinds of negative action tendencies can occur. For example, individuals could be inclined to engage in actions that promote ingroup favoritism and not outgroup derogation, as they are (to some degree) independent of each other (Brewer, 1999). 

2. This alternative hypothesis is easier to generate because the manipulation and the DVs are so similar. The authors told participants that the average response to the question “Blacks and Whites are competing with each other” was either “completely” or “not at all” (i.e. telling participants that Black and White people have terrible or great relations with one another) and then effectively measured perceived relations between the two groups. Here is where not asking about the participants’ beliefs is a hindrance to testing the actual hypothesis. The “perceived” wording/third-party nature of the questions makes social desirability a concern. 

We appreciate you taking the time to lay out this possible alternative. To clarify, the manipulation presents false information that people in the area either perceive high or low levels of Black-White ingroup competition. However, the manipulation check and outcomes ask whether participants themselves perceive this to be the case, not whether they believe others believe this to be the case. Thus, the manipulation presents information about what others believe, while the scales measure what the participant believes. 

Regarding social desirability, we fully agree that in studies like these, it may be a notable concern (and include a section in in our Limitations to address these concerns and alternative approaches for future studies). In fact, part of the reason we phrased the questions in a mutual/bidirectional sense is to avoid favorable responding (e.g., White individuals responding low on White individuals avoiding Black individuals). We believe the effects observed here were not primarily the result of desirability effects. One reason is that, as you stated, anchors in the manipulation range were either 6.27 or 1.73. One would assume that with desirability biased responding, the responses would have mirrored the manipulation. However, participants responses averaged between 3-4 in the high condition and 2-3 in the low condition. Moreover, while condition was the driving factor that influenced outcomes (which was central to this empirical work), we also observed moderation by the structural variable of racial income inequality. That is, participants’ responses were influenced by the objective White-Black income gap in their area. If desirability was the core driver of effects, one would assume that moderation by such a variable would have had little to no effect (Note: other individual differences such as SDO, Support for Economic Inequality, and System Justification also moderated condition effects. To better communicate the scope of this work, we moved these findings to the Supporting Information file).

Taken together, while we agree that desirability effects may have had some influence (as is the case in most every study with self-report measures), we do not believe these wholly explain the pattern of data observed herein. Moreover, the revised discussion calls attention to desirability effects (pg. 27). 

3. This becomes more important for Study 1 because there are various racial groups present but the study focuses specifically on the Black-White divide. 

Non-White and non-Black participants were not excluded from Study 1 because when analyses were conducted, these participants were erroneously not filtered (when, based on relevance, they could have been omitted). To maintain transparency and statistical power, we kept these participants and included a footnote (see Supporting Information file) stating that if these individuals were removed, the effects remain. 

4. The fact that condition (and not race) is the primary driver of the effects is another reason why I feel the alternative hypothesis – that participants are not actually changing their beliefs but reporting beliefs in line with the information told to them by the researchers – might have some merit. The work that does exist on how White people respond to threat (and experiencing competition is usually threatening) is varied. Sometimes they aggressively respond (think shifting demographics and status threat) while other times they move towards appeasement. The equal movement for anxiety, discrimination, mistrust, and avoidance for Black and White people was very surprising from that perspective.

Regarding condition as a driving factor (and not race), importantly, the manipulation was designed to create stark differences between low-perceived competition information and high-perceived competition information. Given that this was an experiment, the effect sizes seen here are influenced by the artificial nature of experimental designs (Baumeister, 2020). Race, on the other hand, is a demographic variable that, while still demonstrated strong associations, was not directly enhanced or primed. As such, seeing a stronger condition effect (compared to race) is not out of the ordinary. 

We agree with your comment regarding variability in White people’s responses to threat evoked by competition. These comments speak to differences in motivation (e.g., approach vs. avoidance; maintaining advantage vs. avoiding disadvantage), which are areas of inquiry currently under investigation in our lab. In the present work, while participants reported on perceived action tendencies that stem from competition threat, we believe that the manner in which items were phrased blunted the potential for threat. That is, given the mutuality of the items (e.g., “Black and White people try to avoid having conversations with each other”), it is unclear which group is experiencing the brunt of the outcome. Furthermore, while the condition effects were similar for Whites and Blacks (i.e. no interactions with race), we, indeed, observed a race main effect (Black participants reported higher perceptions of negative outcomes) which is consistent with the prominent health, education, and income gaps between Whites and Blacks in American society.

Reviewer #2:

Introduction:

1. However, there are problems with some of the logic behind the hypotheses and the discussion of the relevant theories. For example, it is not clear (from the Introduction) exactly why inequality leads to competition; more specifically, why are Whites competitive with Blacks (is it because they want to increase the advantage that they have; is it guilt-driven in an effort to justify the inequities)? Usually one thinks of competition being heightened by equality—e.g., two good teams competing for first place. If the discrimination is overwhelmingly directed against Blacks, wouldn’t they be likely to be more competitive? On the other hand, competitiveness leading to more outgroup discrimination—for both Blacks and Whites-- is definitely intuitive (as a number of previous studies have shown).

Thank you for highlighting the need to clarify our theorizing. Towards this end, we revised the lead-in (pg. 14) to clarify how we think the racial income gap influences perceived competition. We also added theorizing for why we believe these effects would occur for both White and Black individuals (pgs. 14-15). While we highlight why advantaged and disadvantaged groups would be similarly impacted by our manipulation, the question of why Whites, specifically, might be affected by racial income inequality is beyond the scope of this research. 

On equality and competition, we agree that equality can also increase perceptions of competition, however we believe that the nature of the equality matters a great deal. To illustrate, past work looking at the relationship between inequality and competition demonstrates that inequality (measured using the Gini coefficient) positively predicts perceived competitiveness (Sommet et al., 2019). Similarly, racial income inequality predicts both perceptions of competition (broadly) and intergroup (Black-White) competition, with larger income gaps associated with greater perceptions (Gordils et al., 2020).

As for competition leading to discrimination, we agree with this (and address this in our introduction, pg. 4). This link is intuitive, as you suggest, based on previous research. However, prior to this work, no research had provided experimental evidence that manipulations of perceived Black-White intergroup competition directly impact negative intergroup outcomes.

2. In terms of measures, there is some ambiguity as well. The perceived discrimination measure asked how often people in your zip code had these discriminatory experiences. That is a very different question than one asking how much discrimination the respondent has experienced themselves. One would assume that more egalitarian people would be inclined to: see more discrimination and more inequity in their neighborhoods, and report having experienced less discrimination themselves. A suggestion here: the authors should control for the percentages of the different racial / ethnic groups and also for SES levels in the different zip codes.

We appreciate the comment and suggestion. To begin, we deliberately chose to ask participants about the degree to which they perceive discrimination broadly as opposed to discrimination they themselves experienced (directed towards themselves) for several reasons. First, self-directed discrimination measures would require multiple items: whether people had been targets of discrimination and whether they had been agents of discrimination. Second, being a target of discrimination is strongly associated with race (Greene et al., 2006) and we were interested in how perceptions of intergroup competition impacted both advantaged and disadvantaged groups, and we were concerned that we would encounter a floor effect in a self-directed target of discrimination question for advantaged group members (White individuals). Third, asking participants whether they had been agents of discrimination is problematic. Because of myriad biases including desirability, correspondence, and better-than-average effects to name a few, we suspected participants would not accurately and truthfully report on their discriminatory behaviors. Also, discrimination is illegal in many domains/contexts, so we did not want to ask participants about illegal behaviors. 

Regarding egalitarianism, to your point, in ancillary analyses (see Supporting Information file), we controlled for social dominance orientation (SDO), as we were interested in its moderating role on condition main effects. Part of the scale includes antiegalitarian facets and, thus, SDO is negatively correlated with egalitarianism. When controlling for SDO, all condition and race effects remain, although there is moderation for a subset of the effects, such as perceived discrimination (see Table S3). Directly to your point, low-SDO (i.e. more egalitarian) individuals exhibited a stronger condition effect on discrimination. This suggests that those higher in egalitarianism may, indeed, report higher discrimination.

3. A more concerning issue is that the authors are claiming that they manipulated perceived competition between Blacks and Whites, but the manipulation was heavy-handed enough (mean reported agreement with the statement “In my zip code, it seems that blacks and whites are competing with each other” of 6.27 vs. 1.73 on the 1 – 7 scale—a very large difference) that respondents had to assume that there was more and less discrimination in their neighborhoods. The correlations between perceived competition and the various measures of perceived discrimination attest to this. The latter was, in some sense, almost a logical inference from the information the participants had been provided. Compared with this manipulation (ie, its strength), the respondent reports of discrimination and even of interracial competition are not nearly as pronounced. Also, given the strength of the correlations among the different measures of discrimination, the authors might want to combine them into an index.

Please see our response to Reviewer 1’s comment #2, as we believe our response there largely addresses these concerns. Importantly, our manipulation was deliberate, as the goal was ultimately to shift perceptions of interracial competition to see if said changes impact perceptions of intergroup outcomes.

Regarding the issue of respondents “having to assume” there was more and less discrimination, we would like to unpack that a bit. First, the logic that higher (and lower) competition leads to higher (and lower) discrimination is consistent with our theorizing and findings. As mentioned throughout the introduction, we understand that competition can beget discrimination, but no prior research has demonstrated that experimentally manipulating perceptions of Black-White competition directly changes perceptions of discrimination. On the other hand, if what you mean is that the manipulation may be imposing a demand characteristic, such that participants are simply clicking higher numbers for the high condition and lower numbers for the low, we also believe this is not the focal driver of the effects (please see our response to Reviewer #1’s Comment #2). 

Regarding the correlations, while the correlations amongst the outcome variable are relatively large, the correlations between the manipulation check and the outcome variables at maximum account for 30% of the variance, which demonstrates independence amongst these variables. 

Lastly, regarding creating an index, we agree that these outcome variables are associated, in that they may be capturing some latent aspect of racial tension or intergroup antagonism. However, each outcome variables was selected to map onto different affective processes and motivational action tendencies that group members act on in order to cope with competition (e.g., approach-motivated: discrimination; avoidance-motivated: behavioral avoidance; see Esses et al., 2005; Jost & Banaji, 1994; Pratto & Lemieux, 2001; Tajfel et al., 1971). 

A couple of very minor points:

1. Competition has a number of different connotations (some of which come close to those of the- word conflict).

Competition, indeed, has many connotations across many different contexts. For instance, competition can occur between individuals (interpersonal), within groups (intragroup), or between groups (intergroup) to name a few settings, and conflict between social groups has often been tied to competition for resources (see Esses et al., 2005 for a review). Here, we focus on intergroup competition processes occurring between two salient racial groups in American society. Although beyond the boundaries of the data presented here, it is certainly plausible that perceptions of intergroup, White-Black competition are associated with conflict between those racial groups. The revised manuscript seeks to orient the reader to the competitive context we are examining (pgs. 3-6), and we are careful not to overstep the boundaries of the data and generalize to society-level outcomes (pgs. 26-27). 

2. I would suggest taking out the assessment of the study on p. 23: “..the experimental effects observed here represent a welcome addition to the literature…”

We appreciate the suggestion. The sentence now reads:

“Notably, the vast majority of research to date on these topics has been correlational (Rios et al., 2018), thus the experimental effects observed here confirm many underlying assumptions in models of intergroup competition.”

3. same for the use of the term “causal.”

Respectfully, we have opted to retain the term causal when referring to condition effects because of the experimental nature of this research. Participants were randomly assigned to high- or low-competition conditions. Thus, condition effects can be considered as providing causal evidence. However, we are careful not to refer to the quasi-experimental race effects or any moderation tests as causal. 

Journal Requirements:

We modified the manuscript in accordance with the recommended formatting.

2. Please amend your current ethics statement to confirm that your named institutional review board or ethics committee specifically approved this study.

We added the line “Procedures were approved by an ethics board and participants provided consent online prior to participation.” On Page 8. 

The subset of data is part of a manuscript that is currently under review. Importantly, this work is unpublished, and the scope of it focuses on COVID-19 onset effects, and as a pre-COVID-19 sample, we leveraged a subset of Study 1’s data.

4. Please include captions for your Supporting Information files at the end of your manuscript, and update any in-text citations to match accordingly. 

As per request, we added captions and included a Supporting Information section.

---

## [Decision Letter · Decision Letter 1]

1 Dec 2020

PONE-D-20-22299R1

The effect of perceived interracial competition on psychological outcomes

PLOS ONE

Dear Dr. Gordils,

Thank you for submitting your manuscript to PLOS ONE. After careful consideration, we feel that it has merit but does not fully meet PLOS ONE’s publication criteria as it currently stands. Therefore, we invite you to submit a revised version of the manuscript that addresses the points raised during the review process.

The paper is much improved. I appreciate your attention to the reviewers' and my comments. Reviewer 1 has some minor points of clarification. Please address these comments and revise your manuscript to use bias-free language when referring to racial groups (e.g., Black people, instead of Blacks).

We look forward to receiving your revised manuscript.

Kind regards,

Natalie J. Shook

Academic Editor

PLOS ONE

Reviewers' comments:

Reviewer's Responses to Questions

**Comments to the Author**

1. If the authors have adequately addressed your comments raised in a previous round of review and you feel that this manuscript is now acceptable for publication, you may indicate that here to bypass the “Comments to the Author” section, enter your conflict of interest statement in the “Confidential to Editor” section, and submit your "Accept" recommendation.

Reviewer #1: All comments have been addressed

2. Is the manuscript technically sound, and do the data support the conclusions?

Reviewer #1: Yes

3. Has the statistical analysis been performed appropriately and rigorously? 

Reviewer #1: Yes

4. Have the authors made all data underlying the findings in their manuscript fully available?

Reviewer #1: Yes

5. Is the manuscript presented in an intelligible fashion and written in standard English?

Reviewer #1: Yes

6. Review Comments to the Author

Reviewer #1: (No Response)

7. PLOS authors have the option to publish the peer review history of their article (what does this mean?). If published, this will include your full peer review and any attached files.

Reviewer #1: **Yes: **Sa-kiera T. J. Hudson

---

## [Author Response · Author response to Decision Letter 1]

9 Dec 2020

Natalie J. Shook, PhD 

Academic Editor - PLOS ONE

Dear Dr. Shook, 

Thank you again for allowing us to revise our manuscript, “The effect of perceived interracial competition on psychological outcomes" (PONE-D-20-22299R1). We revised the manuscript in response to your comments, and we believe this version of the manuscript fully addresses each of your comments and concerns. Below is a list of the edits made in response to the comments in the order in which they were received (we have copied and pasted the comments below and provide our response to each).

Editor Comments

Reviewer 1 has some minor points of clarification. Please address these comments and revise your manuscript to use bias-free language when referring to racial groups (e.g., Black people, instead of Blacks). 

We address the comments and concerns of Reviewer #1 below. First, changes have been made to the manuscript to use bias-free language throughout. 

Reviewer #1:

1. On page 4, the authors write: “In response to threats from outgroups, ingroup members exhibit motivation to quell the threats, which can take the form of ingroup favoritism (e.g. entitlement justifications; Jost & Banaji, 1994), outgroup derogation (e.g. discrimination; Pratto & Lemieux, 2001), and behavioral avoidance (Esses et al., 2005).” I was confused with the juxtaposition of behavioral avoidance next to ingroup favoritism and outgroup derogation, as usually the literature compares behaviors that are ingroup versus outgroup focused assuming they encompass the full range of behaviors. Thus, avoiding the outgroup could be due to ingroup favoritism (e.g., wanting to be near similar others) or due to outgroup derogation (e.g., seeing the outgroup as inferior and wanting the “riffraff” segregated”).

a. Similarly, in the next paragraph the authors discuss “discrimination” as an example of outgroup derogation yet ingroup favoritism can also lead to discrimination that has nothing to do with demoting the regard of the outgroup. 

b. I encourage the authors to review the manuscript to ensure that the theories and concepts presented are internally consistent.

We appreciate this comment and apologize for the lack of clarity here. You are, indeed, correct that researchers often focus on action tendencies and perceptions akin to ingroup favoritism and outgroup derogation (or outgroup hate). Regarding avoidance behavior, there is quite a bit of literature that documents avoidance as a behavioral response to competition as well as a motivational profile that impacts competition engagement, and more downstream, performance (see Elliot, 2020; Esses et al., 2005). Whether behavioral avoidance should be binned with ingroup favoritism or outgroup derogation may be up to one’s personal interpretation, however, intergroup relations researchers would argue that avoidance is distinct (separate from ingroup favoritism and outgroup derogation; Esses, et al., 2005). Esses, Dovidio, and colleagues argue for three general strategies that are used to reduce competition, namely “outgroup derogation and discrimination”, “ingroup enhancement and preferential treatment,” and “avoidance.” By no means are we arguing that these strategies are unrelated (as demonstrated by the data, indeed, they are related), but at the very least there is justifiable discourse on the separation of these action tendencies. As such, we strongly believe that the way avoidance is presented is warranted. 

Regarding discrimination, we agree that this section could be worded better. First, discrimination is typically linked to outgroup derogation, and we argue that one form of outgroup derogation is discrimination. We do not state (or believe) that discrimination is exclusively linked to outgroup derogation. As you suggest, there is prominent work that links discrimination to ingroup favoritism (Balliet & De Dreu, 2014; Greenwald & Pettigrew, 2014). As such, we revised this section to state the following:

“In response to threats from outgroups, ingroup members exhibit motivation to quell the threats, which can take the form of ingroup favoritism (Jost & Banaji, 1994), outgroup derogation (Pratto & Lemieux, 2001), and behavioral avoidance (Esses et al., 2005). 

Discrimination, or biased treatment of a group or its members (Dovidio et al., 2019), functions to promote positive self-regard in agents by either demoting competing outgroups (i.e., outgroup derogation; Hewstone et al., 2002; Tajfel & Turner, 1986) or by reserving benefits and favors for ingroups (i.e. ingroup favoritism; Brewer, 2017; Hamley et al., 2020).”

2. The authors judiciously use italics, and it becomes difficult to understand what exactly the formatting technique is highlighting.

Per this suggestion, the revised manuscript significantly reduced the use of italics.

3. As I mentioned in the last review, the authors write “perceived interracial competition is associated with lower levels of support for affirmative action programs, higher levels of racial bias and stereotyping, ingroup favoritism, outgroup derogation, and perceived intergroup discord” but it wasn’t clear if the authors are arguing that both Black and White individuals show this effect. One suggestion is to only include the effects that are agnostic to the race/status of the individual, which allows the reader to place the lack of racial interactions within the existing literature. Furthermore, it more concretely explains their hypothesis that race would not matter, which was surprising to me as a reader.

We appreciate this comment and your suggested solution. To add some clarity, we added a footnote after the target sentence that states the following:

1 The majority of these studies examined these associations among mostly White (non-Black) samples.

In much of the classic research on intergroup competition, it was typically the case that researchers would focus on just one group (typically White participants) to examine intergroup perceptions. We recognize the blatant limitations here, which is one of the reasons we sought to recruit equal numbers of Black and White participants in Study 2. 

We do want to note that we are by no means saying that race does not matter. In fact, we find reliable and robust race main effects which suggests the contrary. However, in the context of the competition manipulation, which specifically focuses on mutual involvement of both groups (e.g., “competing against each other”), we hypothesized that both groups would exhibit similar condition effects.

4. I found the discussion of the four outcome variables on pages 25-26 useful to include earlier, around page 6, as it explains why these four variables are important to study in the context of perceived racial competition.

We appreciate this feedback, though we believe the current location of this information is ideal, as it addresses the connection between these outcomes and more plausible downstream negative health outcomes. However, to highlight the importance of these variable in the introduction, we added the following on pages 4-5:

Considering the importance of the aforementioned outcomes, as well as their connection with racial disparities (e.g., Gibbons et al., 2004; Kessler et al., 1999; Pascoe & Smart-Richman, 2009), the work presented here measures perceptions of discrimination, behavioral avoidance, intergroup anxiety, and interracial mistrust as the primary outcomes of interest. 

5. I was confused regarding the information in the brackets in the results sections: e.g., “t(839) = 9.58, [.86, 1.31], p < .001, d = .66.” Are they 95% Cis? If so, of what?

 You are correct in that they are 95% Confidence Intervals, specifically of the mean difference. To add some clarity, we provide the following footnote (now footnote #3):

3 Confidence intervals are at the 95% level and reflect the range of the mean difference between conditions.

6. On page 13 the authors write: “On the one hand, because interracial (and intergroup) competition involves participation and engagement from both ingroups and outgroups, Blacks and Whites may be similarly influenced by perceptions of interracial competition. On the other hand, because Blacks experience worse outcomes compared to Whites across numerous social, psychological, and economic indicators, it is possible that perceived interracial competition disproportionally impacts Black participants”. It is also the case that because White individuals have more to lose if they feel competition, they might be disproportionally affected by competitive threats. The authors’ argument starts to include the status/power differential but feels like one side is missing.

 We appreciate this comment. While we believe that Black participants (compared to White participants) are more likely to experience effects (based on the previously mentioned reasons), it is correct that White individuals also have reasons why they may experience more competition. As such, we modified this section to state the following:

On the one hand, because interracial (and intergroup) competition involves participation and engagement from both ingroups and outgroups, Black and White people may be similarly influenced by perceptions of interracial competition. On the other hand, competition effects may differ as a function of racial group membership. That is, because Black people experience worse outcomes compared to White people across numerous social, psychological, and economic indicators, it is possible that perceived interracial competition disproportionally impacts Black individuals. Alternatively, because high-status groups (e.g., White people) have more to lose from shifting status relations (Wilkins et al., 2015), White people may be more likely to be affected by rising perceptions of interracial competition. Past research, however, suggests that White individuals feel less competitive threat from Black individuals compared to other racial groups (Bobo & Hutchings, 1996). Nonetheless, to address lingering questions tied to racial groups’ responses to competition, Study 2 recruited similar numbers of Black and White participants… 

7. Reading the work on pages 14 and 15, I had no idea why we were all of a sudden reading about race-based income inequality until the very last sentence. If possible, think about integrating the overview in the introduction or better introduce the why of including race-based income inequality beyond “One notable dimension in which Blacks and Whites exhibit a large disparity is in income”. 

 We now see the prior lack of transition regarding the inclusion of racial income inequality. To remedy this, we added the following on page 7:

 Finally, given links between competition and income inequality, we also examined the moderating role of local-area racial income gaps. Past work has posited race-based income inequality may make resource and group differences salient, exacerbating perceptions of competition, both inter-individually and between Black and White racial groups (Gordils et al., 2020; Sommet et al., 2019). Thus, we tested whether objective, local-area racial income inequality moderated effects of condition on the four focal outcomes (i.e., perceived discrimination, avoidance, anxiety, and mistrust). 

8. Is it possible that you found “The effects of condition on these outcomes were stronger for those living in areas with larger racial income inequality” because larger racial income inequality in one’s area made the manipulation more believable? If not, it would be useful to have a sentence or two debunking this and any other alternative hypotheses.

 We appreciate this thoughtful suggestion. We would argue that it is indeed possible that objective levels of racial income inequality make the manipulation more believable, although we cannot conclude that this is the only route through which the local area impacts perceptions. It is also possible that the status of the local environment allows for information regarding interracial competition to become more easily assessible within one’s cognitive/associative network, or perhaps larger gaps in one’s area allow for individuals to be attuned or sensitive to intergroup competition. Why and how racial income inequality moderates condition effects are important questions, though they are outside the scope of the current manuscript.

9. The authors bounce back and forth between active and passive voice. Not sure if it was intentional. As someone that tries to write in active voice but always slips into the passive, I wanted to flag in case that was something the authors wanted to address.

 Thank you for highlighting this. Apart from a few instances (which are now corrected in the revised manuscript), the use of passive voice was deliberate.

---

## [Editor Report · Decision Letter 2]

6 Jan 2021

The effect of perceived interracial competition on psychological outcomes

PONE-D-20-22299R2

Dear Dr. Gordils,

We’re pleased to inform you that your manuscript has been judged scientifically suitable for publication and will be formally accepted for publication once it meets all outstanding technical requirements.

Kind regards,

Natalie J. Shook

Academic Editor

PLOS ONE
---

## [Editor Report · Acceptance letter]

19 Jan 2021

PONE-D-20-22299R2 

The effect of perceived interracial competition on psychological outcomes 

Dear Dr. Gordils:

I'm pleased to inform you that your manuscript has been deemed suitable for publication in PLOS ONE. Congratulations! Your manuscript is now with our production department. 

Kind regards, 

on behalf of

Dr. Natalie J. Shook 

Academic Editor

PLOS ONE